# Anatomical Description of Loggerhead Turtle (*Caretta caretta*) and Green Iguana (*Iguana iguana*) Skull by Three-Dimensional Computed Tomography Reconstruction and Maximum Intensity Projection Images

**DOI:** 10.3390/ani13040621

**Published:** 2023-02-10

**Authors:** Jose Raduan Jaber Mohamad, Eligia González-Rodríguez, Alberto Arencibia, Soraya Déniz, Conrado Carrascosa, Mario Encinoso

**Affiliations:** 1Department of Morphology, Veterinary Faculty, University of Las Palmas de Gran Canaria, Trasmontaña, Arucas, 35413 Las Palmas, Spain; 2Veterinary Teaching Hospital, University of Las Palmas de Gran Canaria, Trasmontaña, Arucas, 35413 Las Palmas, Spain

**Keywords:** computed tomography, maximum intensity projection, volume rendering, reptiles, anatomy, skull, lizard, turtle

## Abstract

**Simple Summary:**

Due to the growing interest in reptiles and the fact that variation in reptile skull anatomy remains poorly documented, we compared the heads of two reptile species using computed tomography reconstructive procedures, such as maximum intensity projections (MIP) and volume rendering (VR). The resulting images demonstrated that these procedures are suitable for performing comparative anatomical studies as they provide adequate information about the different bones that comprise the head while avoiding the overlapping of other structures. A detailed description of the morphology of the head and associated structures could expand the knowledge of clinicians for the diagnosis of traumatic lesions with small or large fractures, skull malformations, and osteodystrophy secondary to nutrition imbalances or neoplasms.

**Abstract:**

The growing interest in reptiles has posed a challenge to veterinary clinicians due to the lack of a standardized system to perform anatomical studies similar to those used for dogs and cats. In this paper, we have attempted to describe, employing computed tomography and subsequent three-dimensional reconstructions, the normal anatomical features that comprise the skulls of two species of reptiles: the loggerhead turtle (*Caretta caretta*) and the green iguana (*Iguana iguana*). Computed tomography (CT) and subsequent image processing allowed the identification of the bony structures that comprise the head of these species. As a result, and based on previous articles, we propose the most significant anatomical differences and similarities between these species.

## 1. Introduction

Modern diagnostic imaging techniques allow for the creation of three-dimensional representations that can be of considerable value in the dissemination of clinical diagnoses and anatomical studies [1]. Thus, the development of such advanced imaging technologies has improved the quality of anatomical imaging, which enables better evaluation and treatment of different animal diseases.

Therefore, these imaging modalities provide highly reliable diagnostic images that have enabled advancements in the fields of both human and veterinary medicine by reconstructing axial images [2,3]. *Reconstruction* is widely misused as it refers to a very specific procedure that converts the raw data into an axial image, however, *reformatting* displays the images produced from the original reconstruction process in a different orientation than that which was initially produced [4]. Among these, we highlight multiplanar reconstructions (MPR), maximum and minimum-intensity projections (MIP and MinIP), and volume rendering (VR) [5,6,7,8,9].

MPR produces a reconstruction along the three axes of space. The X-axis constitutes the right-to-left view, the Y-axis corresponds to the craniocaudal view, whereas the Z-axis is the dorsoventral view of the patient under study, resulting in sagittal and coronal views from cross-sectional images [6].

The most commonly used approaches are the MIP and MinIP. MIP is a procedure that uses only the maximum relative value detected along the ray trajectories retained by the computer, showing preferentially osseous structures and high-contrast regions, while other structures of lower attenuation are not well visualized. Due to its ability to retain attenuation information despite thresholds, it is adequate for the identification of simple structures. Nevertheless, it is conditioned by the poor differentiation of overlapping structures, pulse motion artifact, and breathing [7]. On the other hand, the MinIP detects the minimum values along the ray paths in each view, showing better image quality in those structures with low attenuation [8]. In the literature, these techniques have been reported in human medicine to diagnose different disorders, especially neoplastic diseases [10]. In contrast, only a few articles in veterinary medicine have used these imaging procedures [11,12], although this is expected to increase as they are very useful for the visualization of internal structures [13,14].

Another tool is volume rendering (VR), where the voxels that comprise the image can be programmed to be visible or not, to have different colors, and also to have different levels of opacity, which is called “transparency”. VR additionally provides a very useful tool: “spherical clipping volume” or “shutter tool”, which establishes borderlines between voxels that need to remain visible (tissues inside the sphere) and voxels that need to remain invisible (all adjacent structures). Therefore, the volume data can be resampled based on a series of software algorithms, and manually edited to depict the region of interest from surrounding structures. This procedure has been applied in both human and veterinary medicine, as it is a valuable tool, e.g., for surgical planning [15] and the anatomical study of many species, allowing an acute visualization of bone and soft structures [9]. 

Reptiles have gained increasing interest in veterinary medicine. It has motivated veterinary clinicians to understand physiology and anatomy to improve diagnostics. However, to the authors’ knowledge, most of the studies performed on these species have been conducted using dedicated micro-CT scanners, and only a few descriptions have been reported using conventional CT equipment, highlighting those carried out on different species of reptiles, such as the Komodo dragon, the loggerhead turtle, the common tegu, the green iguana, the boa constrictor, and the bearded dragon [16,17,18,19,20,21,22]. Therefore, this paper aims to provide further insight into the comparative anatomical study of the heads of the green iguana and the loggerhead turtle using conventional CT equipment and 3-D reconstructed images. 

## 2. Materials and Methods

### 2.1. Animals

In this study, we studied five adult female loggerhead turtles (*Caretta caretta*), which weighed between 82 and 106 kg and had a carapace length ranging between 78 and 101 cm. This data was essential for confirming that they were adult specimens. Additionally, we included four adult male green iguanas (*Iguana iguana*), which weighed between 3.8 and 4.1 kg and had a length from snout-to-vent ranging between 28 and 45 cm. All these carcasses were collected from the Rancho Texas Lanzarote Park (Lanzarote, Spain). The animals were scanned at the Veterinary Hospital of Las Palmas de Gran Canaria University. No pathological findings were observed during the clinical examination of the head. We performed anatomical measurements of the skulls of both species (Figure 1) Therefore, we measured three head characters: head length (HL), from the tip of the snout to the posterior border of the collar, head width (HW) at the widest point of the head, and head height (HH) at the highest point of the head. In the loggerhead turtle’s head, the HL mean was 11.03 ± 2.1 cm, the HW mean was 7.94 ± 1.02 cm, and the HH mean was 5.29 ± 0.7 cm. Whereas in the green iguana head, the HL mean was 8.08 ± 0.28 cm, the HW mean was 6.83 ± 0.09 cm, and the HH mean was 4.89 ± 0.07 cm.

### 2.2. CT Technique

Transverse CT images were obtained using a 16–slice helical CT scanner (Toshiba Astelion, Toshiba Medical System, Tokyo, Japan). The reptiles were placed symmetrically in ventral recumbency on the CT couch. A standard clinical protocol (120 kVp, 80 mA, 512 × 512 acquisition matrix, 1809 × 858 field of view, a spiral pitch factor of 0.94, and a gantry rotation of 1.5 s) was used to obtain sequential transverse CT images (1 mm thickness). The original transverse data were recorded and transferred to the CT workstation. No CT density or anatomic variations were detected in the heads of the reptiles used in the investigation. In this study, we applied two CT windows by adjusting the window widths (WW) and window levels (WL) to appreciate the CT appearance of the head structures: a bone window setting (WW = 1500; WL = 300) and a soft tissue window setting (WW = 350; WL = 40). The original data were used to generate head VR and MIP reconstructed images after manual editing of the transverse CT images to remove soft tissues using a standard Dicom 3D format (OsiriX MD, Geneva, Switzerland). 

## 3. Results

Different rostral, lateral, ventral, dorsal, and caudal VR and MIP images were obtained for better visualization and description of the different bones comprising the head (dermatocranium, neurocranium, and mandibula) of the turtle and the iguana (Figure 2, Figure 3, Figure 4, Figure 5, Figure 6, Figure 7, Figure 8, Figure 9, Figure 10 and Figure 11). We selected images that better matched each other, providing valuable information.

### 3.1. Dermatocranium

#### 3.1.1. PREMAXILLA (Os Premaxillare)

The premaxillary bones are narrow, showing a V shape. These bones are the most rostral bony structures of the turtle head (Figure 2A and Figure 3, Figure 4 and Figure 5). Its rostral VR image showed the absence of dorsomedial processes in these bones, allowing the joining of the nostrils (*nares*) at the midline. This paired bone articulates laterally with the maxillary bone (Figure 2A, Figure 3, Figure 4 and Figure 5A) and caudoventrally with the anterior end of the vomer (Figure 4). We distinguished a similar configuration in the iguana, where it comprises a tooth-bearing alveolar plate. This bone contacts the nasal bone dorsally (Figure 7, Figure 8 and Figure 10), the maxilla caudolaterally (Figure 7, Figure 8 and Figure 10), and the vomer caudomedially (Figure 10B). 

#### 3.1.2. MAXILLA (Os Maxillare)

The maxilla is a pair of bones that meet each other at the midline under the palatal bones. Therefore, it is located along the lateral edge of the vomer and palatine bones, surrounding the ventral border of the eye orbit (Figure 2). In the rostral and lateral MIP images of the loggerhead turtle (Figure 2B and Figure 3B), this bone meets rostromedially with the premaxillary bone, ventromedially with the vomer and palatine bones, and caudally along the ventral border of the jugal bone (Figure 2 and Figure 3A). Concerning the iguana, this bone arises lateral and ventral to the nasal bone on each side of the skull, contacting it rostrally with the premaxilla (Figure 7 and Figure 8), the lacrimal and nasal bones dorsally, and the jugal caudolaterally (Figure 8). 

#### 3.1.3. NASAL (Os Nasale)

In the iguana, the nasal bones are broad and slightly convex dorsally. The VR and MIP images showed the nasal bone junction in the mid-sagittal plane and how the nasal bones were distinctly longer than wide (Figure 8), contacting with the neighboring bones such as the lacrimal bone caudolaterally, the maxilla rostrolaterally, the prefrontal bone caudodorsolaterally, and the frontal bone caudodorsomedially (Figure 7, Figure 8 and Figure 10). 

#### 3.1.4. VOMER 

The vomer is an unpaired laminar bone divided into a dorsal and a ventral portion. This last is roughly rectangular and could be distinguished in ventral VR and MIP reconstructions. The vomer is located in the anteromedial palatal region, separating the two external nostril openings (*Apertura narium externa*). The anterodorsal section of this bone forms part of the floor of the nasal passages (*fossa nasalis*), which is well seen with both VR and MIP ventral reconstructions. In the turtle, this bone contacts rostrally with the premaxilla and the maxilla, laterally with the palatine, and caudally with the pterygoid (Figure 4). Referring to the iguana, the vomer is a paired bone with a triangular shape that articulates with the maxilla (rostrally), the palatine bones (caudally) (Figure 9), and medially with one another. Interestingly, this bone shows little or no contact with the premaxilla and none with the pterygoid bone.

#### 3.1.5. PALATINE (Os Palatinum)

The palatine bone overlays the dorsomedial surface of the maxillary bone (Figure 2B). It is a paired laminar bone between the vomer and pterygoid bones. It shows a ventral concave shape that narrows posteriorly, contacting laterally with the jugal bone and medially with the pterygoid bone. The loggerhead turtle and green iguana showed the same configuration, which was easily seen by the VR and MIP ventral reconstructed images (Figure 3B, Figure 4 and Figure 9).

#### 3.1.6. PREFRONTAL-FRONTAL

In these species, the frontal bone shows a flat shape and lies in the anterior part of the skull (Figure 2, Figure 3, Figure 5, Figure 7, Figure 8 and Figure 10). The rostral VR turtle reconstruction showed a clear view of the prefrontal and frontal bone junctions (caudomedially). Therefore, the frontal bone contacts the prefrontal bone (rostrally), the postorbital bone (laterally), and the parietal bone (caudally) (Figure 2, Figure 3 and Figure 5). As our animals were adults, we could observe how this bone was excluded from the orbit. The VR and MIP images of the green iguana showed that the prefrontal bone contacts rostromedially with the nasal bone, rostrolaterally with the lacrimal bone (paired small bones located in the lateral region of the orbit observed only in the iguana), and caudomedially with the frontal bone (Figure 7, Figure 8 and Figure 10). The frontal bone is broad and forms a major component of the skull roof, contributing to the dorsal border of the orbit. It meets the prefrontal bone rostrolaterally, the postfrontal bone caudolaterally, and the parietal bone caudally (Figure 7, Figure 8 and Figure 10). Laterally, it is differentiated from the postorbital by the postfrontal bone. 

#### 3.1.7. POSTFRONTAL-POSTORBITAL

The postorbital bone forms part of the temporal arch, and most of the posterior margin of the orbit. This bone contacts dorsocaudally with the parietal bone, anterodorsally with the prefrontal bone, and ventrally with the jugal bone (Figure 2, Figure 3 and Figure 5), excluding, therefore, the frontal bone from the orbit. It reaches the squamosal caudally (Figure 3A and Figure 5) and contacts the prefrontal bone rostrally (Figure 2A, Figure 3A and Figure 5). Regarding the iguana, those bones were broad and could be well identified in VR and MIP rostral, dorsal, and lateral images as two different bones. The postfrontal bone arises in the caudodorsal part of the orbit and meets the frontal bone (rostromedially), the parietal bone (caudomedially), and the postorbital bone (caudolaterally) (Figure 7, Figure 8 and Figure 10). In contrast, the postorbital bone meets the postfrontal bone dorsally, the squamosal bone caudolaterally, and the jugal bone rostrolaterally (Figure 8 and Figure 10). 

#### 3.1.8. PARIETAL (Os Parietale)

The parietal bone in loggerhead turtles is wide and relatively flat. This bone is the largest component of the skull roof (Figure 5A) and is formed by a horizontal dorsal plate comprising part of the temporal skull roof and a vertical ventral plate (*processus inferior parietalis*) parasagittally (Figure 2, Figure 3, Figure 5 and Figure 6). Moreover, this bone separated the cranial cavity (*cavum cranii*) from the temporal fossa (*fossa temporalis) (*observed in caudal VR view). It contacts the postorbital bone (laterorostrally) (Figure 2, Figure 3B and Figure 5), the frontal bone (mid-rostrally) (Figure 2, Figure 3B and Figure 5), the supraoccipital bone (caudally), and the squamosal bone (caudolaterally) (Figure 3B and Figure 5A). Similarly, the parietal bone in the green iguana is a short, plate-shaped bone with a descending portion that forms part of the temporal fossa (Figure 7, Figure 8 and Figure 10). It was clearly seen using both MIP and VR tools. This bone meets the postfrontal bone (rostrolaterally) (Figure 7, Figure 8 and Figure 10), the squamosal and the supratemporal (laterally) (Figure 8), and the supraoccipital (caudomedially) (Figure 10 and Figure 11). 

#### 3.1.9. JUGAL (Os Jugale)

The jugal bone is one of the main elements of the zygomatic arch under the orbit in both species, showing an “L” shaped configuration in VR and MIP lateral images. Rostral and lateral VR images were quite useful in distinguishing the arrangement of this bone at the caudoventral border of the orbit. Moreover, the rostral, lateral, and ventral VR and MIP images of the turtle show how this bone contacts dorsocaudally with the postorbital bone, rostrally with the maxilla, and posteriorly with the quadratojugal bone (resulting in the zygomatic arch). In contrast, the jugal bone in the iguana is observed as a curved bone (seen by rostral, lateral, and dorsal VR and MIP images) that runs forward along the orbital border to make contact with the lacrimal (dorsally) and maxillary bones rostrally. Caudodorsally, it angles slightly upwards to meet with the postorbital and the squamosal bones. Ventrally, it contacts the ectopterygoid bone.

#### 3.1.10. QUADRATE (Os Quadratum)

The quadrate is a conch-shaped bone and is part of the craniomandibular joint, forming the *fossa temporalis* in both species. The quadrate bone of the turtle contacts dorsally with squamosal (Figure 3 and Figure 4), rostrally is sutured to the pterygoid by the quadratopterygoid process (Figure 4A), and rostrolaterally to the jugal bone by the quadratojugal bone (Figure 3A). In the iguana, this bone contacts dorsally with the supratemporal and the caudal edge of the squamosal bone (Figure 8 and Figure 10). Ventrally, it reaches the articular bone forming the craniomandibular joint (Figure 8).

#### 3.1.11. QUADRATOJUGAL (Os Quadratojugale)

The quadratojugal bone was only observed in the turtle’s head and forms part of the anteroventral border of the cavum tympani and half of the cheek emargination. They are well seen in the lateral VR image. This roughly square bone joins the jugal anteriorly, the postorbital anterodorsally, the squamosal caudodorsally, and the quadrate bone caudally (Figure 3A). 

#### 3.1.12. SQUAMOSAL (Os Squamosum)

The squamosal bone is located posterodorsally above the quadrate and the quadratojugal, and both form part of the antrum postoticum, where the quadrate bone comprises the base and the squamosal the roof in these species. This bone, in the turtle, is attached to the caudal region of the quadrate bone (Figure 3A), dorsally joins the parietal bone (Figure 5A), caudomedially contacts with the paraoccipital process of the opisthotic bone (Figure 6), rostrolaterally with the quadratojugal (Figure 3A), and ventrally with the basioccipital bone (Figure 4). In the iguana, the squamosal bone is a roughly L-shaped bone that shares the same configuration as in the turtle and is well visualized using both VR and MIP techniques in lateral (Figure 8), dorsal (Figure 10), or caudal images (Figure 11). Nonetheless, in the iguana, we could distinguish the presence of a large ventral process of the squamosal that is offset from the main axis of the bone.

#### 3.1.13. PTERYGOID (Os Pterygoideum)

This flat bone lies between the vomer and palatine bones (rostrally) and the parabasisphenoid (caudally) bones (Figure 4) in the loggerhead turtle, reaching the quadrate bone caudally and the palatine bone rostrally (Figure 3B and Figure 4). Interestingly, this bone is fussed with the turtle. In contrast, the iguana was completely separated (Figure 9B,C). This bone has a dorsal projection called the epipterygoid that contacts the parietal bone (Figure 3B and Figure 8) in both species. Nonetheless, we observed that the iguana showed two rostral projections called ectopterygoid bones (Figure 8 and Figure 9), which contacted rostromedially with the pterygoid palatine process (Figure 9B,C), and laterally, it contacts with the jugal and the maxillary bones (Figure 8 and Figure 10). 

#### 3.1.14. EPIPTERYGOID

The epipterygoid bone in the turtle consists of a tubular-shaped bone that extends vertically from the pterygoid bone and contacts the parietal (Figure 3B). A similar configuration was observed in the iguana, where this bone is a thin road that touches dorsally the parietal bone (Figure 8).

### 3.2. Neurocranium

#### 3.2.1. PARABASISPHENOID-BASISPHENOID (Os Basisphenoidale)

The parabasisphenoid is a roughly trapezoidal bone located between the pterygoid (rostrally), and the basioccipital (caudally) bones (Figure 4, Figure 5B, Figure 6, Figure 8 and Figure 9). This bone forms part of the ventral surface of the skull and has two lateral processes that contact the pterygoid bone, called basipterygoid processes, which are short and thick, and well identified in the iguana (Figure 9C). Between these processes, we observe a thin and tubular structure named the cultriform process (Figure 8 and Figure 9), which is well developed compared to other iguana species. This latter also shows two alar processes in the laterocaudal direction (Figure 9B,C). 

#### 3.2.2. BASIOCCIPITAL (Os Basioccipitale)

The basioccipital bone in the turtle is a rectangular-shaped bone located in the posterior half of the planum basale, contacting the basisphenoid (rostrally) and exoccipital (caudally) bones, forming part of the floor of the *cavum cranii* (Figure 3B, Figure 4 and Figure 6). In the iguana, it roughly reaches the parabasisphenoid and caudally the occipital bone (Figure 11), forming part of the occipital condyle.

#### 3.2.3. SUPRAOCCIPITAL (Os Supraoccipitale)

The supraoccipital bone is an unpaired midline element located in the posterodorsal portion of the skull (Figure 3, Figure 4, Figure 5, Figure 6, Figure 10 and Figure 11). Both turtles and iguanas have a similar configuration. This bone has a broad contact with the posteromedial margins of the parietals (Figure 3B, Figure 5A, Figure 10 and Figure 11). Posteroventrally, it contacts the opisthotic and otoocipital, forming part of the medial part of the occipital condyle, near the foramen magnum (Figure 6 and Figure 11). These features were observed by VR and MIP reconstructions in the caudal view of both species. 

#### 3.2.4. EXOCCIPITAL (Os Exoccipitale)

This bone is a paired bone that lies lateral to the foramen magnum and comprises part of the occipital condyle. Dorsally, it is in contact with the supraoccipital, anterodorsolaterally with the opisthotic bone, and ventromedially with the basioccipital bone (Figure 4A, Figure 5A and Figure 6). The exoccipital forms the posterior border of the metotic fissure. The ossification extends ventrally to the dorsolateral aspect of the occipital condyles, where it joins the basioccipital bone (Figure 4A). 

### 3.3. Mandible 

In both species, the mandible is very robust and shows two pieces that are fused cranially at the intermandibular symphysis.

#### 3.3.1. DENTARY (Os Dentale)

The dentary bone is the major bone of the mandible in both species. Its lateral surface appears pierced by small foramina, where mandibular nerve branches and blood vessels run. In the iguana, it carries pleurodont teeth. This bone was better visualized in the rostral, lateral, and ventral VR and MIP images. Either in the turtle or the iguana, it joins the coronoid and the surangular caudodorsally (Figure 2, Figure 3, Figure 4, Figure 8 and Figure 9), the angular caudoventrally (Figure 3, Figure 4, Figure 8 and Figure 9), and the splenial bone medially. 

#### 3.3.2. ANGULAR (Os Angulare)

The angular bone is a laminar bone located in the caudal aspect of the mandible from the rostromedial to the caudolateral aspect, forming the ventrocaudal margin of the mandible. This bone was well visualized in the lateral MIP and VR images of these species (Figure 3 and Figure 8). The connections of the angular bone in both species are the dentary bone rostrally, the splenial and prearticular bones dorsomedially, and the articular bone in the caudal direction. 

#### 3.3.3. SURANGULAR (Os Surangulare)

The surangular is a widely flat bone located below the coronoid along the dorsal border of the mandible and forming most of the caudolateral surface of the mandible. In both species, it is located posteriorly and slightly dorsal to the dentary, connected to the coronoid rostrodorsally, to the angular bone caudoventrally, and to the articular bone medially (Figure 3 and Figure 8). 

#### 3.3.4. CORONOID (Os Coronoideum)

The coronoid is a broadly triangular bony structure located approximately medial to each branch on the caudodorsal aspect of the mandible. This process is the highest part of the mandible in both species. This bone was better visualized in the lateral MIP view of the iguana than in the turtle, where we could distinguish how the coronoid tends to be slightly higher and more dorsally pointed than in the loggerhead turtle. Both species have the same configuration where the coronoid meets the dentary rostrolaterally, the surangular caudolaterally, the articular caudoventromedially, and the splenial bone ventromedially (Figure 3B, Figure 7 and Figure 8).

#### 3.3.5. PREARTICULAR (Os Prearticulare)

The prearticular bone is a wide, flat lamella overlying a large area of the caudomedial surface of the lower jaw, forming a long, narrow process whose dorsal border constitutes the medial border of the adductor fossa. This bone reaches the coronoid rostrodorsally, the articular bone caudomedially, and the angular ventrally (Figure 3). If the splenial bone is present, this bone is sutured to the rostral edge of the prearticular bone.

#### 3.3.6. ARTICULAR (Os Articulare)

The articular bone is an irregularly block-shaped bone found at the caudal edge of the mandible in the turtle and the iguana. It was observed using both VR and MIP techniques (Figure 3, Figure 6, Figure 7, Figure 8, Figure 9, Figure 10 and Figure 11) and had the same configuration in both species. Thus, it joins with the surangular laterally and the angular ventrally, forming the retroarticular process, and dorsally with the coronoid bone. 

#### 3.3.7. HYOID APPARATUS (Os Hyoideum)

The hyoid apparatus is located in the intermandibular space. It provides support to the tongue, pharynx, as well as floor of the mouth. This apparatus consists of a pentagonal piece, which has three structures: a central (basihyal) structure, an anterior process, and two caudolateral processes (Figure 3A, Figure 4, Figure 5, Figure 6A, Figure 8, Figure 9, Figure 10A and Figure 11). 

## 4. Discussion

In this study, we have used different imaging techniques to evaluate the skulls of two species of reptiles. The results of this investigation have demonstrated that MIP and VR images of the turtle and iguana skulls provided adequate information and understanding of the different bones that comprise the skull. Nonetheless, although a direct comparison between VR and MIP images was not possible, a remarkable correlation in the matched images was obtained. As in other reptiles, the heads of the loggerhead turtle and the green iguana are highly complex structures composed of different bones with significant differences [17,18]. Our results have demonstrated that the imaging reconstruction techniques used in our study were valuable in visualizing these differences. Compared to other studies performed on reptiles, such as the boa constrictors [21], the green iguana, the common tegu, and the bearded dragon that used conventional radiography [22], the acquisition of reconstructed images of the skull reduced the superimposition of the bilateral structures of the snout and the neurocranium, providing better visualization of the configuration of the bones that make up the skull of these species. 

MIP is a volume rendering procedure that projects a volume of interest onto a 3-dimensional viewing plane based on high-intensity structures on CT [9]. Thus, the ability of MIP images to preserve attenuation information makes them widely applicable for visualizing areas of interest or specific anatomic structures. Hence, the images obtained in our study allowed excellent visualization of the external and internal structures that concern the skull of both species. Therefore, we identified specific bones of the dermatocranium as the premaxilla, the maxilla, the nasal, the vomer, and the palatine. In the neurocranium, we distinguished dorsal and lateral bones as the frontal, the parietal, and the different parts of the pterygoid, as well as their configuration and relation with other bones, such as the epipterygoid and the parietal bones. However, we could not identify other head bones, such as the basisphenoid, the supraoccipital, or the dentary, which have already been identified using micro-computed tomography [23]. This specific equipment provides higher spatial resolution and thinner slice thickness than that obtained with conventional CT scanners; therefore, micro-CT scanners provide standard images and three-dimensional renderings of extraordinary quality and a superior level of detail compared with standard CT units [24,25]. Nonetheless, micro-CT units are not usually available in veterinary hospitals because of the limited range of species that can fit into the gantry [26].

Three-dimensional volume rendering (VR) is a powerful imaging tool that reconstructs 2D CT data into a 3D model with high fidelity [9]. The main advantage is that it facilitates a three-dimensional visualization, avoiding the superimposition of surrounding structures and therefore allowing for the identification of internal structures [9,16,19]. The VR images allowed us to see how the head of the loggerhead turtle was wider in the posterior part and narrower before the orbits, and the jaw was robust with a fused symphysis. These features have been associated with greater biting performance than those of other sea turtle species [27]. Despite these findings, the skull of *Caretta caretta* shares similar features to those of other sea turtles, such as large orbits and a short and blunt rostrum [23,28]. Concerning the orbit, VR and MPI were quite helpful in visualizing how the frontals were excluded from the border of the orbit by the prefrontal bones. A similar finding was identified in other studies using micro-CT [23]. This orbital disposition is not observed in other species of sea turtles, such as the Kemp’s ridley. The skull of the iguana is convex with a short preorbital region but a similar V-shaped jaw configuration. Concerning internal bones, VR ventral images allowed the depiction of different configurations of the pterygoid bone in both species. As stated in other studies performed on sea turtles [23,28], the pterygoid bone of the loggerhead turtle shows prominent lateral processes. In contrast, in the iguana, this bone was separated from the bone on the other side. Some authors have reported an age-dependent degree of ossification of the pterygoid bone, especially at the pterygoid-epipterygoid junction, tending to fuse with age [29]. However, with both techniques, we could observe that this fusion did not occur in the iguana. Other valuable structures of the head identified by VR were the scleral ossifications, which have been studied by several researchers as a possible growth marker, although scleral ossicles would not be an alternative for loggerhead sea turtle skeletochronology. However, they could be helpful for other sea turtle species [30]. Additionally, this technique allowed us to ascertain the differences in the quadrate bone in both species. Therefore, VR lateral images served to visualize the relation between other bones and the quadratojugal bone and to confirm the absence of this last bone in the iguana head. Similar findings were also described in other studies [31,32,33], which confirmed that the presence of quadratojugal is a misinterpretation. In contrast, VR and MIP lateral images were helpful in identifying the adductor fossa in the iguana’s head. The presence of this chamber in the iguana has been proposed as indicative of the development of jaw adductor muscles in this species [34].

In conclusion, these two techniques were quite helpful, providing anatomical references for the bone structures that comprise the heads of these species. Compared to the excellent resolution provided by micro-CT, VR combined with MPI provides adequate detail of the bone structures that compose the head. Therefore, the results obtained in this study can be adequate to evaluate numerous processes, such as traumatic lesions with small or large fractures involving the skull that can produce brain damage, skull malformations, and osteodystrophy secondary to nutrient imbalances or neoplasms. Moreover, these two reconstruction techniques can facilitate teaching veterinary anatomy to students by realistically allowing the view of structures without superimposition of other structures, eliminating the difficulties of visualizing the extension of different types of lesions.

## Figures and Tables

**Figure 1 animals-13-00621-f001:**
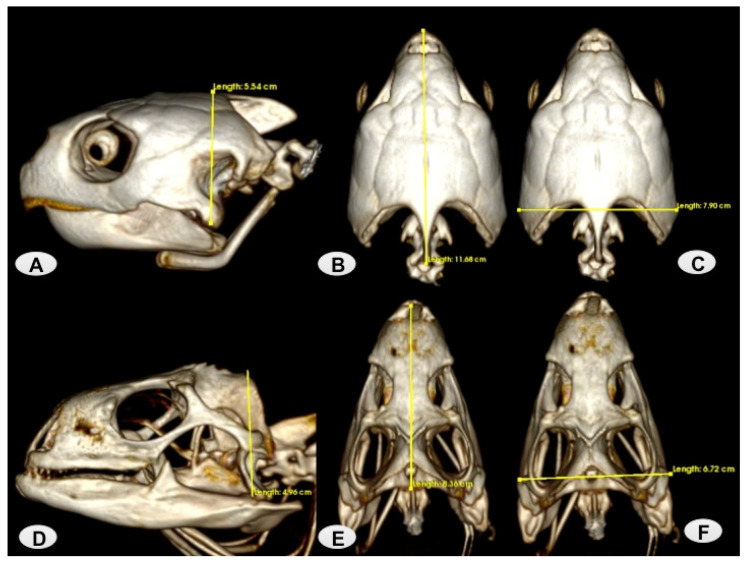
Measurements of the loggerhead turtle and green iguana Head. (**A**) Example of HH in a loggerhead turtle lateral VR image. (**B**,**C**) Examples of HL and HW in loggerhead dorsal VR images. (**D**) Example of HH in a green iguana lateral VR image. (**E**,**F**) Examples of HL and HW in green iguana dorsal VR images.

**Figure 2 animals-13-00621-f002:**
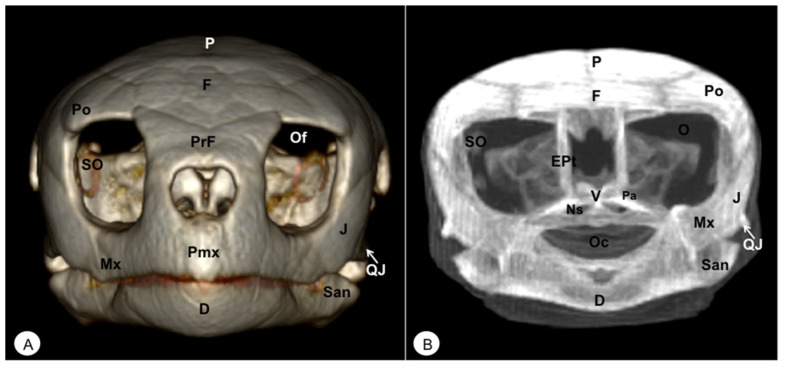
Head of *Caretta caretta*. (**A**) Rostral VR image. (**B**) Rostrotransversal MIP image. Pmx: Premaxillary bone. PrF: Prefrontal bone. F: Frontal bone. Po: Postorbital bone. O: Orbit. Of: Orbital fossa. J: Jugal bone. QJ: Quadratojugal. Mx: Maxillary bone. SO: Scleral ossifications. P: Parietal bone. EPt: Epipterygoid bone. V: Vomer. Pa: Palatine bone. Ns: Nasal sinuses. Oc: Oral cavity. D: Dentary. San: Surangular.

**Figure 3 animals-13-00621-f003:**
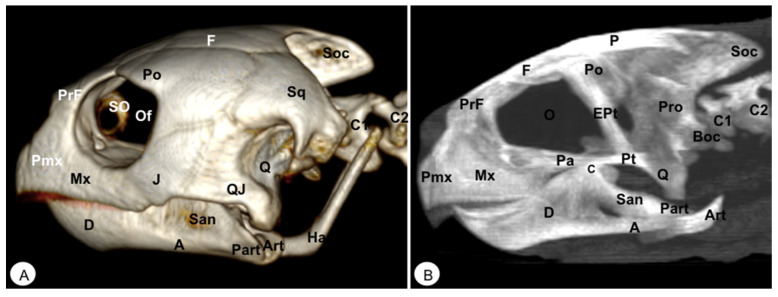
Head of *Caretta caretta*. (**A**) Lateral VR image. (**B**) Medial MIP image. Pmx: Premaxillary bone. Mx: Maxillary bone. PrF. Prefrontal bone. F: Frontal. Po: Postorbital. J: Jugal. SO: Scleral ossifications. QJ: Quadratojugal. Q: Quadrate. Sq: Squamosal. Boc: Basioccipital. Pt: Pterygoid. EPt: Epipterygoid. Pa: Palatine. P: Parietal. Soc: Supraoccipital. Pro: Prootic. C: Coronoid. O: Orbit. Of: Orbital fossa. D: Dentary. A: Angular. San: Surangular. Part: Prearticular. Art: Articular. Ha: Hyoid apparatus. C1: First cervical vertebra. C2: Second cervical vertebra.

**Figure 4 animals-13-00621-f004:**
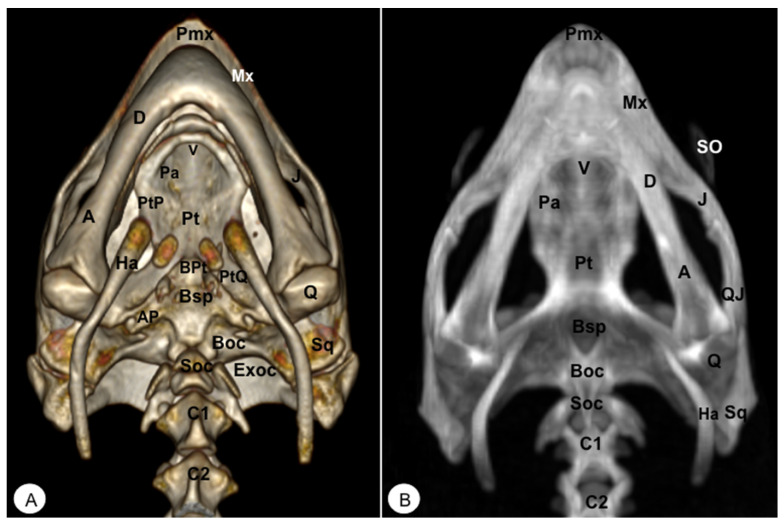
Head of *Caretta caretta*. (**A**) Ventral VR image. (**B**) Ventral MIP image. Pmx: Premaxillary bone. Mx: Maxillary bone. J: Jugal. V: Vomer. Pa: Palatine. PtP: Pterygoid palatine process. BPt: Basipterygoid. Pt: Pterygoid. PtQ: Pterygoid quadrate process. SO: Scleral ossifications. Sq: Squamosal. Q: Quadrate. QJ: Quadratojugal. Bsp: Basisphenoid. AP: Articular process of Bsp. Boc: Basioccipital. Soc: Supraoccipital. Exoc: Exoccipital. M: Mandible. D: Dentary. A: Angular. Ha: Hyoid apparatus. C1: First cervical vertebra. C2: Second cervical vertebra.

**Figure 5 animals-13-00621-f005:**
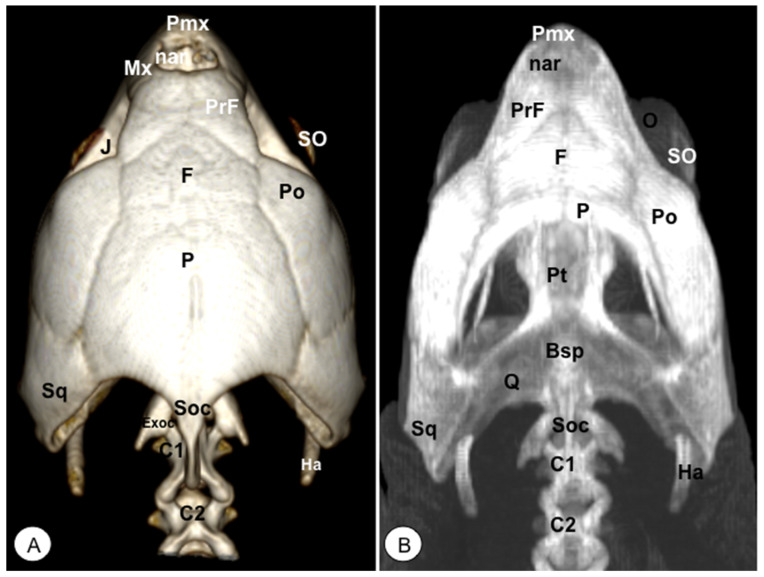
Head of *Caretta caretta*. (**A**) Dorsal VR image. (**B**) Dorsal MIP. Pmx: Premaxillary bone. Mx: Maxillary bone. nar: Nares. PrF: Prefrontal. F: Frontal. Po: Postorbital. J: Jugal. O: Orbit. SO: Scleral ossifications. Sq: Squamosal. Pt: Pterygoid. Bsp: Basisphenoid. P: Parietal. Exoc: Exoccipital. Soc: Supraoccipital. Ha: Hyoid apparatus. C1: First cervical vertebra. C2: Second cervical vertebra.

**Figure 6 animals-13-00621-f006:**
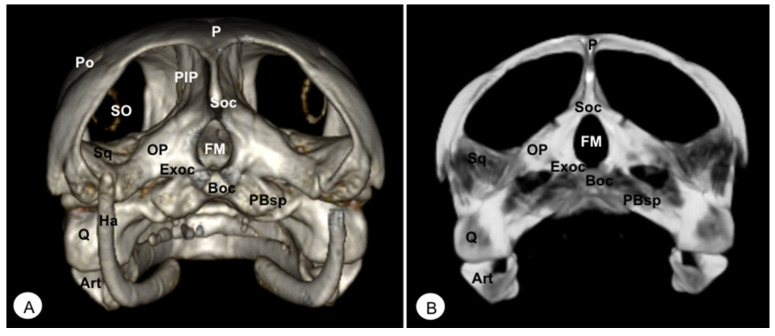
Head of *Caretta caretta*. (**A**) Caudal VR image. (**B**) Caudal MIP image. P: Parietal bone. PIP: Procesus inferior parietalis. Po: Postorbital. Sq: Squamosal bone. Soc: Supraoccipital bone. Exoc: Exoccipital bone. Boc: Basioccipital bone. OP: Opisthotic bone. FM: Foramen magnum. PBsp: Parabasisphenoid bone. SO: Scleral ossifications. Q: Quadrate. Art: Articular. Ha: Hyoid apparatus.

**Figure 7 animals-13-00621-f007:**
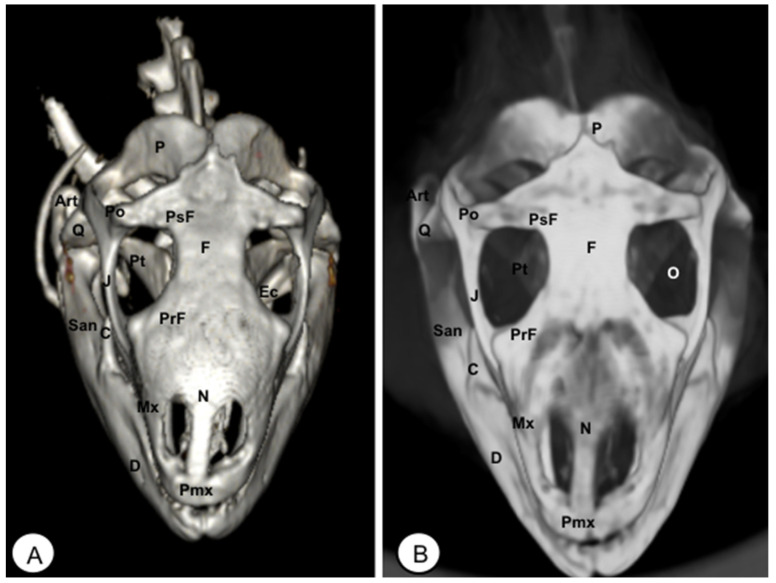
Head of *Iguana iguana*. (**A**) Rostral VR image. (**B**) Rotral MIP image. N: Nasal. Pmx: Premaxillary bone. Mx: Maxillary bone. PrF: Prefrontal. F: Frontal. PsF: Postfrontal. Po: Postorbital. J: Jugal. Q: Quadrate. P: Parietal. O: Orbit. Pt: Pterygoid. Ec: Ectopterygoid. Art: Articular. C: Coronoid. D: Dentary. San: Surangular.

**Figure 8 animals-13-00621-f008:**
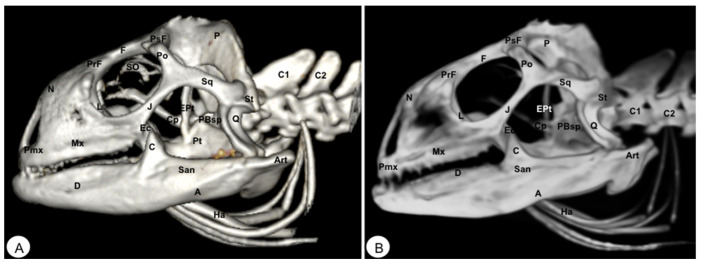
Head of *Iguana iguana*. (**A**) Lateral VR image. (**B**) Lateral MIP image. Pmx: Premaxillary. Mx: Maxillary bone. N: Nasal. PrF: Prefrontal. L: Lacrimal. F: Frontal. PsF: Postfrontal. Po: Postorbital. SO: Scleral ossifications. J: Jugal. Q: Quadrate. Sq: Squamosal. Ec: Ectopterygoid. Pt: Pterygoid. EPt: Epipterygoid. Cp: Cultriform process. PBsp: Parabasisphenoid. P: Parietal. St: Supratemporal. D: Dentary. A: Angular. San: Surangular. Art: Articular. C: Coronoid. Ha: Hyoid apparatus. C1: First cervical vertebra. C2: Second cervical vertebra.

**Figure 9 animals-13-00621-f009:**
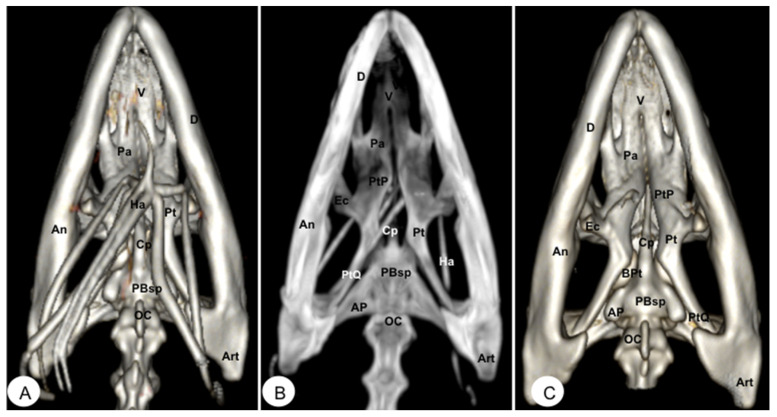
Head of *Iguana iguana.* (**A**) Ventral VR image. (**B**) Ventral MIP image. (**C**) Ventral VR image without Hyoid apparatus. D: Dentary bone. An: Angular bone. Art: Articular bone. V Vomer. Pa: Palatine bone. Pt: Pterygoid bone. Ec: Ectopterygoid. PtP: Pterygoid palatine process. PtQ: Pterygoid quadrate process. BPt: Basipterygoid process. PBsp: Parabasisphenoid bone. Cp: Cultriform process. OC: Occipital condyle. Ha: Hyoid apparatus. AP: Alar process.

**Figure 10 animals-13-00621-f010:**
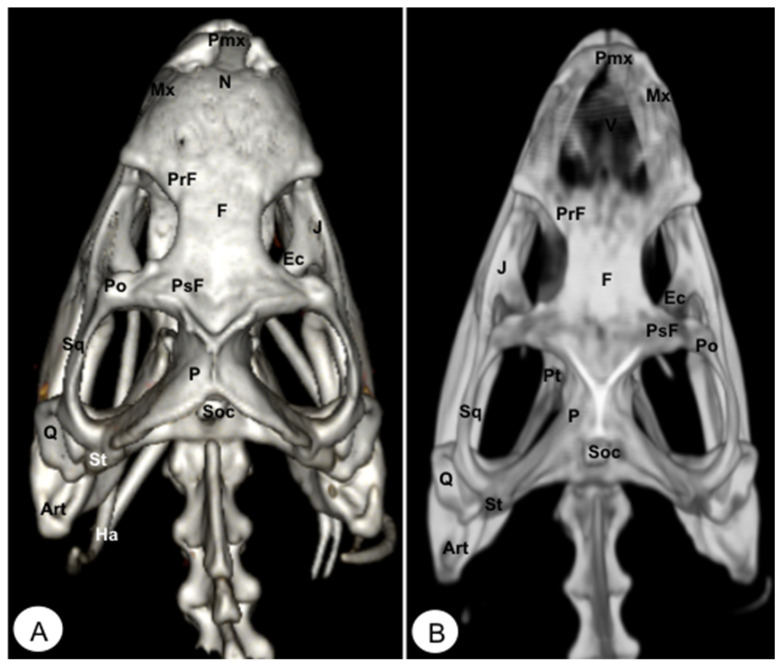
Head of *Iguana iguana*. (**A**) Dorsal VR view. (**B**) Dorsal MIP view. Pmx: Premaxillary bone. Mx: Maxilla. N: Nasal. PrF: Prefrontal. F: Frontal. PsF: Postfrontal. Po: Postorbital. J: Jugal. Pt: Pterygoid. Ec: Ectopterygoid. Q: Quadrate. Sq: Squamousal. St: Supratemporal. P: Parietal. Soc: Supraoccipital. Art: Articular. Ha: Hyoid apparatus.

**Figure 11 animals-13-00621-f011:**
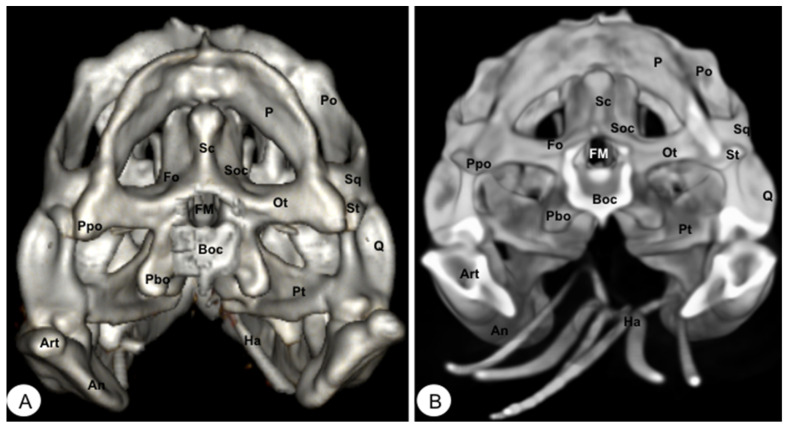
Head of *Iguana iguana*. (**A**) Caudal VR image without hyoid apparatus. (**B**) Caudal MIP image. Sc: Occipital sagital crest. Soc: Supraoccipital. P: Parietal. Ot: Otoccipital. FM: Foramen magnum. Pt: Pterygoid. Po: Postorbital. Sq: Squamosal. St: Supratemporal. Boc: Basioccipital. Ppo: Paroccipital process. Pbo: Parabasioccipital process. Fo: Otoccipital facet. An: Angular. Art: Articular. Ha: Hyoid apparatus.

## Data Availability

Not applicable.

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
