# Peer review of "Anatomical Description of Loggerhead Turtle (Caretta caretta) and Green Iguana (Iguana iguana) Skull by Three-Dimensional Computed Tomography Reconstruction and Maximum Intensity Projection Images"

_animals, 2023, doi:10.3390/ani13040621_

Round 1

Reviewer 1 Report (Previous Reviewer 2)

The present version of the manuscript has been substantially improovred. The numbers of experimental animals significantly increased.

However, some points need to be clarified:

1.      Affilaitions should be written in English, please correct 22 in realtion to Conrado Carrascosa

2.      Details of experimental animals are still lacking (age, weight).

3.      Were animals anestethised or not ?

4.      Still do not know why the authors did not decide to measure any of cranium bones. It would be of benefit.

Author Response

Reviewer 1 comments for Author:

  1. Affiliations should be written in English, please correct 22 in relation to Conrado Carrascosa

We have corrected all the affiliations. Therefore, in this revised version, all of these ones are written in English. Besides, we have changed that referred to Conrado Carrascosa.

  1. Details of experimental animals are still lacking (age, weight). 

The details of the experimental animals are included in the material and methods, within the animal section. Therefore, we have included “In this study, we studied five adult female loggerhead turtles (Caretta caretta), which weighed between 82-106 kg and with a carapace length ranging between 78-101 cm. This data was essential to confirm that they were adult specimens. Besides, we included four adult male green iguanas (Iguana iguana), which weighed between 3.8-4.1 kg and with a length from snout-to-vent ranging between 28-45 cm.”

  1. Were animals anaesthetized or not?

No, they were not since all these animals were carcasses, collected from the Rancho Texas Lanzarote Park (Lanzarote, Spain).

  1. Still do not know why the authors did not decide to measure any of the cranium bones. It would be of benefit.

We completely agree with this comment. Therefore, measuring the different skull bones comprising the cranium would be very important. Although the segmentation of the individual bones that compose the neurocranium and dermatocranium was not the purpose of our work, the measurements we obtained in 3D VR of a single bone of the skull were not accurate. Thus, we did not include those measurements, although, in further studies, we will follow your advice and use specific software to reach this purpose.

Reviewer 2 Report (New Reviewer)

Anatomical work using modern imaging technologies on reptiles is rare. Hence, the research topic undertaken is most appropriate.

The study was performed on five adult female loggerhead turtles (Caretta caretta) and four adult male green iguanas (Iguana iguana). This is sufficient material. The research techniques used are state-of-the-art and provide valuable information, both cognitive and clinically applicable. I rate the quality of the illustrations included very highly. The literature cited is appropriate and represents the state of the knowledge.

Accordingly, please add Latin names in the titles of the subsections of section 3.1 Dermatocranium, e.g:
 31.3 Nasal (os nasale)
Palatine (os palatinum)
Do the same for subsection 3.2 Neurocranium.

The authors juggle anatomical nomenclature very freely. Sometimes they use Latin names, other times they do not. Please apply the rule that if a structure is mentioned for the first time, we put the Latin name in brackets.

[403] please remove the words deleted and highlighted in green

[303] Fig 8 not Fig 7.

Author Response

Reviewer 2 comments for Author:

Accordingly, please add Latin names in the titles of the subsections of section 3.1 Dermatocranium, e.g:

31.3 Nasal (os nasale)

Palatine (os palatinum)

As you suggested, we have added Latin names in all the subsection titles

The authors juggle anatomical nomenclature very freely. Sometimes they use Latin names, other times they do not. Please apply the rule that if a structure is mentioned for the first time, we put the Latin name in brackets.

As you suggested, we have corrected the anatomical nomenclature along with the results. Therefore, we put the Latin name of those structures in brackets.

[403] please remove the words deleted and highlighted them in green

Following your suggestion, we have removed the words highlighted in green.

[303] Fig 8, not Fig 7.

This suggestion has been corrected.

Round 2

Reviewer 2 Report (New Reviewer)

The authors have made adjustments to make the paper easier to read on the one hand and to bring it into order on the other. Minor inaccuracies in figure numbering have also been corrected. I believe that the paper in its current version is of great cognitive and clinical interest.

This manuscript is a resubmission of an earlier submission. The following is a list of the peer review reports and author responses from that submission.

Round 1

Reviewer 1 Report

This manuscript attempts to use skulls of Iguana iguana and Caretta caretta as examples to showcase a method of CT scanning. However, as currently written and presented, it is well below industry standard in a number of aspects. A significant amount of work is required to improve it for it to be a worthwhile publication. Below I provide a list of numbered general issues and a list of specific suggestions. I hope that the authors find these helpful and constructive. As  is convension, I would expect an itemised and duely considered reply in return.  

GENERAL ISSUES

1. Novelty - method

CT scanning is not a new technique. There have been detailed descriptions of reptiles using X-ray CT scanning for more than 15 years (and much longer for mammals). Perhaps the authors are using some new software or application which is easy and quick to use. However, they are not explicit about which software and do not make this point clear. Very few publications about CT scanning methods are cited to put their methods in the context of those described elsewhere. There are many articles out there that could be mentioned to clarify what they have done, e.g.

- Carlson, W.D., Rowe, T., Ketcham, R.A. and Colbert, M.W., 2003. Applications of high-resolution X-ray computed tomography in petrology, meteoritics and palaeontology. Geological Society, London, Special Publications, 215(1), pp.7-22.

- Kleinteich, T., Beckmann, F., Herzen, J., Summers, A.P. and Haas, A., 2008, September. Applying X-ray tomography in the field of vertebrate biology: form, function, and evolution of the skull of caecilians (Lissamphibia: Gymnophiona). In Developments in X-Ray Tomography VI (Vol. 7078, pp. 113-122). SPIE.

- Spoor, F., Jeffery, N. and Zonneveld, F., 2000. Using diagnostic radiology in human evolutionary studies. The Journal of Anatomy, 197(1), pp.61-76.

2. Novely - taxon choice

Iguana iguana and Caretta caretta seem bizzare choices to me given that both species are relatively well well known anatomically having already been described in the literature in great detail dozens of times. Both species have appeared in text books (e.g. Walker, W.F. and Liem, K.F., 1994. Functional anatomy of the vertebratesan evolutionary perspective.). The senior author of the submission is even the senior author on a paper (Pérez et al. 2021) which includes similar data on Iguana iguana. 

3. Scholarship

Although Iguana and Caretta are not novel taxa to describe the authors could have turned this weakness into something of a strength by using the previous literature to 

- illustrate how important both taxa have been as comparative taxa to date for understanding evolution, function, development, allometry, or interpretation of fossils, 

- thoroughly scrutinising the variation evident among previous descriptions of these two taxa that appear to be in conflict,

- evalutate their own description in this context. 

However, the authors have made no real attempt to do any of this work. Moreover, as written, this manuscript makes the authors look extremely uninformed with respect to the literature and internet. 

Key descriptions of the skull of Iguana iguana that NEED to be acknowledged:

- Bochaton, C., Grouard, S., Breuil, M., Ineich, I., Tresset, A. and Bailon, S., 2016. Osteological differentiation of the Iguana Laurenti, 1768 (Squamata: Iguanidae) species: Iguana iguana (Linnaeus, 1758) and Iguana delicatissima Laurenti, 1768, with some comments on their hybrids. Journal of herpetology, 50(2), pp.295-305.

- Conrad, J.L. and Norell, M.A., 2010. Cranial autapomorphies in two species of Iguana (Iguanidae: Squamata). Journal of Herpetology, 44(2), pp.307-312.

- Evans, S.E., 2008. The skull of lizards and tuatara. Biology of the Reptilia, 20, pp.1-347.

You should probably also acknowledge and link to:

https://people.ohio.edu/witmerl/3D_iguana.htm 

https://sketchfab.com/3d-models/green-iguana-skull-ouvc-10677-1427f887e0fd41eb9c9c97e07c3610e3 

https://www.youtube.com/watch?v=lka9Qszhyds

Key descriptions on the skull of Caretta caretta that NEED to be acknowledged:

- Chatterji, R.M., Hipsley, C.A., Sherratt, E., Hutchinson, M.N. and Jones, M.E., 2022. Ontogenetic allometry underlies trophic diversity in sea turtles (Chelonioidea). Evolutionary Ecology, pp.1-30.

- Chatterji, R.M., Hutchinson, M.N. and Jones, M.E., 2021. Redescription of the skull of the Australian flatback sea turtle, Natator depressus, provides new morphological evidence for phylogenetic relationships among sea turtles (Chelonioidea). Zoological Journal of the Linnean Society, 191(4), pp.1090-1113.

- Gaffney ES. 1979. Comparative cranial morphology of recent and fossil turtles. Bull Am Mus Nat Hist 164: 67–376.

- Hay, O.P., 1908. On three existing species of sea-turtles, one of them (Caretta remivaga) new. Proceedings of the United States National Museum.

- Jones, M.E., Werneburg, I., Curtis, N., Penrose, R., O’Higgins, P., Fagan, M.J. and Evans, S.E., 2012. The head and neck anatomy of sea turtles (Cryptodira: Chelonioidea) and skull shape in Testudines. Plos one, 7(11), p.e47852.

- Lunardon, E.A., Costa-Schmidt, L.E., Lenz, A.J., Borges-Martins, M. and de Oliveira, L.R., 2020. Skull ontogenetic variation of the coastal developmental stage of the loggerhead turtle (Caretta caretta) in the western South Atlantic Ocean. Hydrobiologia, 847(9), pp.1999-2019.

- Wyneken, J. and Witherington, D., 2001. The Anatomy of Sea Turtles. NOAA Technical Memorandum NMFS-SEFSC-470. NOAA, Miami, FL, pp.1-172.

You may also want to acknowledge:

http://digimorph.org/specimens/Caretta_caretta/

But *please* ignore the Reptile Evolution website run by David Peters. 

Given that you mention "diagnosis of fractures, metabolic alterations and neoplastic processes." in the abstract you should probably mention a relevant example in the introduction, e.g.: 

- Anderson, M.P. and Capen, C.C., 1976. Nutritional osteodystrophy in captive green iguanas (Iguana iguana). Virchows Archiv B, 21(1), pp.229-247.

- Franchini, D., Cavaliere, L., Valastro, C., Carnevali, F., Van Der Esch, A., Lai, O. and Di Bello, A., 2016. Management of severe head injury with brain exposure in three loggerhead sea turtles Caretta caretta. Diseases of Aquatic Organisms, 119(2), pp.145-152.

4. Quality of description

The descriptions of each bone are reasonably well written and correct but it just don't go far enough. Many modern reptile anatomy descriptions segment individual bones out, describe bone overlaps, and label foramina particularly when these reveal useful distinctions between species and general (e.g. Bell et al. 2003; Bever et al 2005; Klembara et al. 2017). The descriptions provided here are very general and could apply to just about any reptile skull. e.g. "The maxilla is a paired bone located along the lateral edge of the vomer and palatine bones, surrounding the ventral border of the eye orbit" Sure that's true and a good start but so what? 

I appreciate that not every lab can afford to purchase expensive software such as Amira, Avizo, or Mimics for segmenting individual bones. However, software such as SPIERS is free: 

https://spiers-software.org/

Alternatively the authros could still attempt some comparative anatomy with the data they have and previousl descriptions. Please consider adding a couple of sentences after each bone description stating how your skull bones differ or resemble other descriptions of Iguana and Caretta present in the literature. Can you find any within species variaton with respect to suture junctions etc.? 

Examples of excellent descriptions of other taxa for you to get ideas from:  

Bever, G.S., Bell, C.J. and Maisano, J.A., 2005. The ossified braincase and cephalic osteoderms of Shinisaurus crocodilurus (Squamata, Shinisauridae). Palaeontologia Electronica, 8(1), pp.1-36.

Klembara, J., Dobiašová, K., Hain, M. and Yaryhin, O., 2017. Skull anatomy and ontogeny of legless lizard Pseudopus apodus (Pallas, 1775): heterochronic influences on form. The Anatomical Record, 300(3), pp.460-502.

4.1
The submitted manuscript fails to mention that in Caretta caretta the maxillae meet each other at the midline under the palatal bones - It is a key feature of this taxon(!) which has been known for a long time, e.g.
- Jones, M.E., Werneburg, I., Curtis, N., Penrose, R., O’Higgins, P., Fagan, M.J. and Evans, S.E., 2012. The head and neck anatomy of sea turtles (Cryptodira: Chelonioidea) and skull shape in Testudines. Plos one, 7(11), p.e47852.
- Frazier J (1985) Misidentifications of Sea Turtles in the East Pacific: Caretta caretta and Lepidochelys olivacea. J Herpetol 19: 1–11.

4.2
The submitted manuscript also refers to not identifying the the quadratojugal in Iguana iguana (line 396) as if the authors expected to find it. In reality it has long been known that the quadratojugal is not present in Iguana iguana (or indeed any squamate, please see see Evans 2008). Please clarify this text with citations to avoid misleading the reader. 

4.3
The opening labelled in 7A is the lower temporal fenestra or inferior temporal fenestra. It is not the adductor fossa. Fossa means gutter or trench and it sometimes used for a depression around a fenestra but it should not be used for the fenestra itself.   

4.4
Caretta caretta does not have nasal bones (please see Jones et al. 2012; Chatterji et al. 2022). This is a widely documented trait among reptile skull anatomy literature. 

5. Quality of the images

The images are not bad but include errors and could be lablled in ways that would be a lot more valuable to the reader. 

5.1
As mentioned above Caretta does not have nasal bones. You need to relabel figure 1, 2, 3, and 4. 

5.2
The boundary between bones, where one bone ends and another bone begins is frequently unclear (e.g. jugal and quadratojugal). Do not be afraid to add a grey line, or dotted line, to make bone-bone relationships more clear. Do not be afraid to add a ? if you are uncertain. Also in Fig.2 the "pmx" has been placed on the maxilla. Please recheck all labelling and try to clarify it.

6. Method - Animals

How were these animals obtaines? where are they from? Have they been deposited in a museum where they can be examined by others? 

7. Data sharing and open science

The authors do not appear to be making their data available online (e.g. via Digimorph, Morphomuseum, or Morphosource). This practice is a requirement for pubishing descriptions in many serious journals and is viewed as good practice. 

SPECIFIC SUGGESTIONS

Line 13

Is companion animals really the best phrase to use here? 

Line 13

The wording "lack of studies to understand their anatomy?" implies that there are hardly anay when in reality there are many studies out there on reptile anatomy and that includes ones that use X-ray CT. A search in google scholar for "reptile anatomy" provides over 40,000 hits and this number grows every day. You could argue that variation in reptile skull anatomy remainly poorly documented but that is different to the current wording. Moreover, Iguana and Caretta are two of the best known reptiles with repect to reptile anatomy so choosing these two taxa seems to undermine this set up. 

Line 18

Why not be more specific and state soft tissues such as scales, skin, ligaments, and muscles?

Line 36

Which software?

Line 37 

Software such as?

Line 73

This phrase "companion" animals seems rather odd in the context of loggerhead turtles. Perhaps you should provide more explanation as what you mean here. 

Line 170

"Similarly" is probably a better word than "likewise".

Line 122

Caretta caretta does not have nasal bones. 

line 360

You should cite some specific articles to support your statement. You can't simply state "Compared to other works that used conventional radiography, the acquisition of recon- structed images of the skull reduced the superimposition of the bilateral structures of the snout and the neurocranium, providing better visualization of the configuration of the bones that make up the skull of these species." without citing any examples. 

Line 373

Regarding "In contrast, we did not identify other bones of the head such as the basisphenoid, the supraoccipital or the dentary by MPI.", other studies have been able to achieve this feat using x-ray CT, e.g. Jones etal. 2012. 

Line 396

Why would you expect to find the Quadratojugal in an Iguana iguana?

Line 410 Acknowledgements

The current text is fine but should anything else be added regarding access to the CT scanner or specimen material?

Reviewer 2 Report

In general, the concept of this may be really interesting (if done correctly). But in the present form there are many methodological shortcomings which unable to publish the manuscript.  I reject this manuscript because of the too low numbers of animals used.

Line 45 - MPR has already been abbreviated in line 42.

Line 49, 56 – please use MinIP and MPR as previously abbreviated in line 43.

Line 82 - the issue how to establish the minimal number of animals to test to get valuable  data is really disputable. Although, there are some directions provided by Universities or Journals (https://www.nature.com/articles/laban0508-193a) most of researchers agree that depending on kind of study/experiment the minimal number of experimental animals should be 5 or 6 (I personally agree with this). Thus, n=2 is definitely too small.

Line 82 – details of experimental animals are also lacking (age, sex, weight, etc.).

Line 101 – the authors provided basic anatomical description of the cranium bones but did not measure any of them. It is obvious limitation of the study. Additionally, such kind of generalization is risky, especially since the time (age) factor  has not been taken into account.

Line 308 – “Vental” or ventral?

Line 352 – The discussion rather poorly put the authors’ observations it the light of the previous anatomical descriptions of the skull of others reptiles.

Line 401 – the conclusion in present form confirms rather lack of novelty and scientific value of this study.